# Development and Evaluation of a Multifrequency Ultrafast Doppler Spectral Analysis (MFUDSA) Algorithm for Wall Shear Stress Measurement: A Simulation and In Vitro Study

**DOI:** 10.3390/diagnostics13111872

**Published:** 2023-05-27

**Authors:** Andrew J. Malone, Seán Cournane, Izabela Naydenova, James F. Meaney, Andrew J. Fagan, Jacinta E. Browne

**Affiliations:** 1School of Physics, Clinical and Optometric Sciences, IEO Centre, Faculty of Science and Health, Technological University Dublin, D07 H6K8 Dublin, Ireland; 2Tissue Engineering Research Group (TERG), Department of Anatomy, Royal College of Surgeons in Ireland, D02 YN77 Dublin, Ireland; 3Medical Physics and Clinical Engineering Department, St Vincent’s Hospital, D04 T6F4 Dublin, Ireland; 4National Centre for Advanced Medical Imaging (CAMI), St James Hospital and with the School of Medicine, Trinity College Dublin, D08 NHY1 Dublin, Ireland; 5Department of Radiology, Mayo Clinic, Rochester, MN 55902, USA

**Keywords:** flow phantom, biomedical signal processing, cardiovascular disease, Doppler ultrasound, quantitative ultrasound, signal processing algorithms, ultrasonography, wall shear stress

## Abstract

Cardiovascular pathology is the leading cause of death and disability in the Western world, and current diagnostic testing usually evaluates the anatomy of the vessel to determine if the vessel contains blockages and plaques. However, there is a growing school of thought that other measures, such as wall shear stress, provide more useful information for earlier diagnosis and prediction of atherosclerotic related disease compared to pulsed-wave Doppler ultrasound, magnetic resonance angiography, or computed tomography angiography. A novel algorithm for quantifying wall shear stress (WSS) in atherosclerotic plaque using diagnostic ultrasound imaging, called Multifrequency ultrafast Doppler spectral analysis (MFUDSA), is presented. The development of this algorithm is presented, in addition to its optimisation using simulation studies and in-vitro experiments with flow phantoms approximating the early stages of cardiovascular disease. The presented algorithm is compared with commonly used WSS assessment methods, such as standard PW Doppler, Ultrafast Doppler, and Parabolic Doppler, as well as plane-wave Doppler. Compared to an equivalent processing architecture with one-dimensional Fourier analysis, the MFUDSA algorithm provided an increase in signal-to-noise ratio (SNR) by a factor of 4–8 and an increase in velocity resolution by a factor of 1.10–1.35. The results indicated that MFUDSA outperformed the others, with significant differences detected between the typical WSS values of moderate disease progression (*p* = 0.003) and severe disease progression (*p* = 0.001). The algorithm demonstrated an improved performance for the assessment of WSS and has potential to provide an earlier diagnosis of cardiovascular disease than current techniques allow.

## 1. Introduction

Cardiovascular disease (CVD) is the leading cause of death in the world [1]. CVD is typically characterised by a number of biological processes, such as intimal thickening in the vessel wall, development of atherosclerotic plaque, and the stiffening of arterial walls. It has been reported that a number of life-limiting and life-threatening conditions are associated with atypical mechanical features of arterial tissue [2,3]. There is a clinical interest in each of these features, particularly arterial wall stiffening, which has been associated with endothelial damage, leading to CVD progression [4]. The current gold standard diagnostic techniques for CVD include digital subtraction angiography, computed tomography angiography, magnetic resonance angiography, and duplex ultrasound [5,6,7]. Each of these techniques rely on the examination of already present arterial stenoses, limiting their applicability to mid- and late-stage CVD, for which the treatment options are mostly surgical and carry inherent risk [5].

An alternative approach is to measure the endothelial damage that precedes the development of atherosclerotic plaque by detecting changes in the mechanical properties of the vessel wall. One possible biomarker of this damage, wall shear stress (WSS), has been linked to CVD for many years, with several major studies having examined its role in atherogenesis. WSS has been shown to be particularly useful in the assessment of vascular reactivity [8], wall thickening [9], and atherosclerosis [10,11]. In the 1970s, it was postulated that a high WSS value was responsible for the development of CVD. However, more recent research has determined that it is, in fact, low WSS values which are the primary indicators of future CVD [11]. Low WSS values have also consistently been associated with increased risk of saccular aneurysm rupture [12].

Particle image velocimetry (PIV) [13,14,15,16] and phase-contrast magnetic resonance imaging (PC-MRI) [17,18,19,20] have been used to measure WSS; however, the low penetration depth of PIV and the long scan times and high cost of PC-MRI limit their suitability and accessibility as a screening tool. Pulsed-Wave (PW) Doppler ultrasound has been identified as potentially providing a good diagnostic basis for WSS assessment given its low cost and high spatial resolution [21,22]. Furthermore, the temporal resolution of PW Doppler ultrasound can be improved through the use of “UltraFast” plane wave Doppler ultrasound [23]. Ultrafast ultrasound provides both an improvement to the temporal resolution and an improvement to the signal-to-noise ratio (SNR) [24]. There remains, however, a need to further improve both the SNR and the velocity resolution of Doppler velocity quantification in order for ultrasound evaluation of WSS to become a viable diagnostic option. 

Loupas and Gill [25] first proposed the spectral analysis technique of multifrequency Doppler, which allows for decreased Doppler spectral variance and an increase in SNR. This is achieved by deriving spectral information from additional frequencies transmitted in any finite pulse. Although originally developed in spectral Doppler, the multifrequency Doppler technique was mainly used for two-dimensional autocorrelation as the computational requirements for full two-dimensional spectral Doppler could not be met until recently [26,27].

The purpose of this work was, thus, to develop and evaluate a novel signal processing algorithm, which could combine the temporal resolution improvements of ultrafast Doppler with advanced 2-dimensional Fourier signal processing techniques towards the improvement of WSS quantification. 

## 2. Materials and Methods

The Multifrequency Ultrafast Doppler Spectral Analysis (MFUDSA) algorithm was developed, expanding on principles first described in a Doppler analysis technique reported previously [25], and optimised using a combination of simulations and in-vitro phantom studies through an iterative process as described in the following sections. 

### 2.1. MFUDSA Algorithm Development

The data used for the analysis in this work were acquired using a dedicated research package installed on the Aixplorer ultrasound scanner (Supersonic Imagine, Aix-en-Provence, France), which allowed extraction of raw in-phase quadrature (IQ) data. A flow chart of the steps used in the MFUDSA algorithm is included in Figure 1. The processing was performed off-line using code developed in MATLAB^®^ (Mathworks, Natick, MA, USA).

The algorithm initially performed two operations in the time domain. Firstly, the region of interest was defined manually, which allowed the algorithm to discard unnecessary field of view data outside the vessel, resulting in significant memory savings. This step also separated the data into a series of pulse lines, which were analogous to pulsed wave (PW) Doppler pulses. Secondly, a Bartlett windowing function was applied to the time domain, a standard computation often performed in signal analysis before implementing a Fourier transform. The purpose of this step was to truncate the signal close to the start and at the endpoints, as including them would have led to the formation of sidelobe artefacts in the frequency domain. A two-dimensional discrete Fourier transform was then performed, generating a frequency space bounded by the transmitted radiofrequency (MHz) and the Doppler shift frequency (kHz). Once in the frequency domain, a low pass filter was applied to the data to remove the pulse repetition frequency. 

A spectral scaling was then applied to the data, the purpose of which was to “realign” the spectrum, such that integration would correctly sum the individual spectra. The limits of integration for the spectral averaging were taken as the transmitted bandwidth of the transducer, measured using a needle hydrophone, as previously described [28]. Using this measured pulse length, a scaling factor could be applied, such that a frequency could be scaled according to the ratio f_RF_/f_c_. Spectral averaging was applied using the equation developed by Loupas and Gill [25]:(1)$${\hat{P}}_{MF}\left(f_{d}\right)=\frac{\int_{f_{c}-BW/2}^{f_{c}+BW/2}P(f_{RF},\frac{f_{d}f_{RF}}{f_{c}})df_{RF}}{\int_{f_{c}-BW/2}^{f_{c}+BW/2}{\left|U(f_{RF}-f_{c})\right|}^{2}df_{RF}}$$
where ${\hat{P}}_{MF}\left(f_{d}\right)$ is the two-dimensional Doppler spectrum, f_c_ is the central transmitted frequency, BW is the bandwidth of the pulse, ${\left|U(f_{RF}-f_{c})\right|}^{2}$ is the Fourier transform of the pulses complex envelope, and $P(f_{RF},\frac{f_{d}f_{RF}}{f_{c}})$ is the two-dimensional frequency spectrum with the spectral scaling applied. The output from the multifrequency Doppler computation would be equivalent to a time-averaged Doppler frequency histogram across the entire acquisition time. 

Loupas and Gill specified a so-called quality factor for the performance of the multifrequency Doppler computation.
(2)$$Q=\frac{f_{c}}{BW\sqrt{ln(2)}}$$
where f_c_ is the central transmitted frequency, and BW is the pulse bandwidth. It was hypothesised that the quality factor would have an inverse relationship with SNR [25]. In order to compare directly with the results of that earlier study, the Q factor of each pulse length setting was calculated. 

### 2.2. MFUDSA Algorithm Optimisation

As part of the optimisation procedure for the MFUDSA algorithm, the response of the algorithm was evaluated as transmission parameters were varied. The transmission parameters included pulse length, transmission frequency, data length, and pulse repetition frequency (PRF). The impact of the transmission parameters values on the SNR and velocity resolution for the MFUDSA algorithm were evaluated using a hydrophone system, described previously by Ivory et al. [29], as well as an anatomically realistic flow phantom, previously reported by Malone et al. [30].

### 2.3. MFUDSA Algorithm Evaluation

The MFUDSA algorithm was initially tested using two simulated datasets: the first featured a linearly increasing Doppler frequency, corresponding to a velocity range from −20 to +20 cm s^−1^, while the second had a Doppler shift varying as a cosine wave with peak Doppler shifts corresponding to velocities of −40 to +40 cm s^−1^, simulated to feature the reflections of a bandwidth of transmitted frequencies in order to test the multifrequency analysis of the algorithm. 

In vitro testing involved the use of both a string phantom and anatomically realistic flow phantom. Data acquired using a Doppler string phantom (Model 043, CIRS Inc., Norfolk, VA, USA), using a pre-generated paediatric umbilical blood flow pulsatile waveform, and compared with the simulated data. This waveform was selected due to its peak velocity of 45 cm s^−1^, which closely resembles the renal artery peak velocity of ~50 cm s^−1^ [31]. The string phantom data were acquired using the research mode of the Aixplorer scanner, with a transmit frequency of 5 MHz, a pulse length of 4 half cycles, and a PRF of 4 kHz. The number of data points recorded from the Doppler pulse, the data length, was 2000.

An anatomically realistic flow phantom testbed, based on the renal artery, was developed to mimic the wall stiffening associated with the progression of CVD. Three flow phantoms were produced using polyvinyl alcohol cryogel (PVA-c) [32,33] as a vessel mimicking material and IEC agar [34,35] as a tissue mimicking material. The vessel mimicking material stiffness was characterised using representative samples and were produced to have vessel stiffnesses values of 60 kPa, 110 kPa, and 320 kPa. Full details of the production protocol were outlined in a previous paper [30].

### 2.4. Validation of Algorithm in Comparison to Other Techniques

The performance of the MFUDSA algorithm was compared to three other techniques used for the measurement of WSS. WSS is defined as the product of the dynamic viscosity and the gradient of the blood velocity flow profile evaluated as close to the vessel wall as possible, described by the expression:(3)$$\mathrm{WSS}=-\mu \frac{\delta V}{\delta r}{|}_{r=R}$$
where *µ* is the dynamic viscosity of blood $\frac{\delta V}{\delta r}$ is the velocity flow profile, sometimes referred to as the wall shear rate, and *R* is the vessel radius. 

The first technique used PW Doppler, calculating the WSS from equation 3 using velocity measurements manually acquired by moving the range gate. 

The second technique, also using PW Doppler, was a simplification of the WSS equation based on Poiseuille’s law and the assumption of parabolic flow, hereafter referred to as the parabolic method, expressed as [20]:(4)$$\mathrm{WSS}=\frac{2\mu V_{max}}{R}$$
where *μ* is the dynamic viscosity of blood, *Vmax* is the peak velocity recorded at the centre of the vessel and *R* is the vessel radius. This technique is often used as a method to overcome the need to completely quantify the velocity flow profile of a vessel, instead requiring only a single velocity measurement made in the centre of the vessel. The third technique compared to the MFUDSA algorithm was a standard ultrafast acquisition. In principle, the data measured in ultrafast mode should be identical to that of the PW Doppler, and they were acquired to illustrate that the MFUDSA algorithm is distinct from just using an ultrafast acquisition.

### 2.5. Analysis

For each technique, cross-sectional velocity profiles were determined and used to compute the WSS at longitudinal positions along the vessel. An area of particular interest was a position approximately 10 mm distal to the curvature, which has been noted as the region where 60% of renal artery stenoses form [36]. 

For each vessel stiffness value and WSS estimation technique a graph of WSS with respect to longitudinal vessel position was produced. A sample value was taken at the same vessel position for each dataset, at the position of stenosis formation 10 mm distal to the vessel curvature. For each estimation technique, the differences in WSS values between the three vessel stiffnesses were analysed using a paired *t*-test. A comparison between the low (60 kPa) and intermediate (110 kPa) vessel stiffnesses, and the low (60 kPa) and high (320 kPa) vessel stiffnesses were evaluated, respectively. For each test, the null hypothesis was that there was no significant difference (*p* < 0.05) in WSS values as the vessel stiffness increased, and the alterative hypothesis was that there was a detectable difference in WSS values as the vessel stiffness increased.

## 3. Results

### 3.1. MFUDSA Algorithm Optimisation

Based on the velocity outputs from the one-dimensional and two-dimensional processing, direct comparisons were possible for the parameters of SNR and velocity resolution, which was taken as the full width at half maximum (FWHM) of the main velocity peak.

Figure 2 shows the ratio of the two-dimensional SNR to the one-dimensional SNR, here called R_SNR_. The figure shows that, while there is no obvious value of transducer half cycle, which will maximise the R_SNR_ for all cases, the value does appear to be increased for lower half cycle settings. Figure 3 shows the ratio of 1D FWHM to 2D FWHM, here called R_FWHM_. This figure shows that the value of R_FWHM_ was maximised at four transducer half cycles for all cases. As such, this value was selected to be used for all further experiments to provide the largest increase in velocity resolution while maximising the improvement in SNR.

It was found that the Q factor was minimised for a transmission frequency of 5 MHz with a pulse length of 4 half-cycles (Q = 1.99). The Q factor was found to vary between 1.99 and 12.13 for all pulse lengths tested.

### 3.2. Simulation Data

The results for the two simulated datasets describing linear and cosine varying velocity signals are shown in Figure 4. The simulated sonograms appeared to mimic the specified waveforms. The sonogram of the Doppler string phantom, using a paediatric umbilical waveform, can be seen in Figure 5a. An additional simulated sonogram is included in Figure 5b, using the same inputs as Figure 4a, but with a data length of 2000, matching that of Figure 5a.

### 3.3. In Vitro Phantom Data

Figure 6a shows the boxplots for each of the techniques used in the steady flow regime, and Figure 6b shows the boxplots for each of the techniques used in the pulsatile flow regimes. The data length of the research mode acquisitions of the Aixplorer was too short to allow for the generation of coherent sonograms using the MFUDSA algorithm. As such, it could not be compared in pulsatile mode to the other techniques analysed.

Figure 6a showed that for each estimation technique the WSS was decreasing with increasing vessel stiffness. The results for steady flow (Table 1) indicated that, for each technique investigated, there was a significant change in WSS values as the vessel stiffness increased except for the parabolic estimation method comparing low and medium stiffnesses. The largest degree of separability was noted for the MFUDSA algorithm, which showed the lowest *p*-values and the least WSS uncertainties.

Figure 6b showed a similar trend across the estimation techniques for pulsatile flow as Figure 6 showed for steady flow, with an evident decrease in WSS with increased vessel stiffness. The results for pulsatile flow in Table 1 indicate that there was no significant difference (*p* < 0.05) between the low and medium vessel stiffnesses, although both PW Doppler and ultrafast Doppler had *p*-values only marginally greater than the 0.05 significance threshold.

## 4. Discussion

The purpose of this work was to develop and evaluate the performance of a novel signal processing algorithm, which combined a novel multifrequency Doppler signal processing algorithm with ultrafast plane wave acquisition technology, with the end goal of providing an improvement in diagnostic quantification of WSS, a clinical biomarker associated with CVD.

In this study, a statistically significant difference in WSS values for different vessel wall stiffnesses was found in vitro, corresponding to different replicated stages of CVD in phantoms. The values recorded within the phantom for the set low wall stiffness value of 60 kPa varied in the range 3.4–6.2 Pa for steady flow and 1.3–2.9 Pa for pulsatile flow. These values fall within the ranges of normative artery WSS values of 1–7 Pa for low risk and −0.4–0.4 Pa for at-risk patients (reported in the comprehensive review article by Malek et al. [11]). The values reported by Malek et al. for atherosclerosis prone flow conditions were −0.8–0.9 Pa for steady flow and 0.1–0.6 Pa for pulsatile flow, which are consistent with the values found in this study. The MFUDSA algorithm demonstrated significantly improved performance compared the other techniques, and, if it is considered alone, the values recorded for the high stiffness vessel of −0.2–0.3 Pa lie precisely within the region of at risk WSS values.

Chatzizisis et al. [37] reported similar values of WSS for regions at risk of developing atherosclerosis to [11], with WSS values above 3 Pa considered to be high and WSS ≤ 1 Pa considered to be low and at risk. Using this metric, the MFUDSA algorithm again performs well, clearly delineating the three stiffness levels with the low stiffness not at risk, the medium stiffness potentially ‘at risk’, and the high stiffness estimated to be at risk. Of note, the maximum values of WSS for the lowest vessel stiffnesses in Figure 6b were less than half those observed in Figure 6a, likely indicating that the flow did not reach a fully parabolic profile for the pulsatile flow regime, resulting in a flatter velocity profile and a lower WSS value.

Additionally, following the optimisation of the MFUDSA algorithm, there was an observed increase in SNR by a factor of four to eight and an increase in velocity resolution by a factor of 1.10–1.35 compared to the equivalent processing architecture with one-dimensional Fourier analysis. The improvements in SNR were found predominately due to the shorter pulse length settings, while the improvements in velocity resolution were optimized at four transducer half cycles. The length of an ultrasound pulse is inversely proportional to the bandwidth of transmitted frequencies [38] and that long pulses with small frequency bandwidths exhibit a corresponding increase in axial and lateral resolution [22,39]. Consequently, the pulse length is also an important parameter for the purposes of two-dimensional Fourier analysis, as it indirectly defines the available extra frequency information used in the spectral averaging procedure of the MFUDSA algorithm. The results of this work are also consistent with results reported by Loupas and Gill [25] for a one-dimensional Fourier analysis, who investigated four values of Q factor: 2.4, 4.8, 9.6, and 19.2, corresponding to pulse bandwidths of f_c_/2, f_c_/4, f_c_/8, and f_c_/16 respectively. The researchers found that the SNR for a one-dimensional Fourier analysis was independent of Q factor, while the SNR measured herein for two-dimensional multifrequency Doppler exhibited an inverse relationship with Q factor (relative SNR improvement of 70–360% for Q factors in the range 2.4–19.2).

It can be seen from Table 1 that, between techniques, there is a difference in the levels of significance, which is most likely due to the flow regime itself, with a pulsatile flow producing a more fluctuating flow profile, with the parabolic technique exhibiting the lowest sensitivity.

There are two key assumptions made in the parabolic assessment method: that there is fully developed, parabolic flow in rigid vessels, and that the liquid behaves as a Newtonian fluid. Neither of these assumptions are valid in all cases when performing measurements in vivo. Another significant shortcoming associated with the parabolic profile assumptions is that the blood velocity profile is often not axisymmetric and fully developed in vivo. This limits its applicability to vessels, which are particularly long and straight, although is it not necessarily true that characteristically long and straight vessels will exhibit fully developed flow [40]. The parabolic method could exhibit further reduced performance with the progression of CVD; Malone et al. [40] demonstrated that, for low-degree (<50%) stenoses, there can be significant skewing of the position of the peak velocity. This skewing can limit the effectiveness of the parabolic technique [41]. A study carried out by Mynard et al. [42] found that measurement of WSS using parabolic velocity profile assumptions could result in errors of 30–60%.

The main benefit to using the parabolic assumptions is to decrease the scanning time over traditional PW Doppler methods, although this comes at the cost of accuracy. With the advent of ultrafast scanning technologies, this trade off may no longer be necessary as the ultrafast mode could complete an insonation faster than either estimation method could be performed when using PW Doppler, with reductions in acquisitions times of up to a factor of 16 [24].

The ultrafast Doppler mode behaved similarly to the PW Doppler mode, with no statistically significant improvement in assessment accuracy or precision. However, the ultrafast mode has several significant benefits compared to PW Doppler, for example it can acquire the same velocity information in a fraction of the time and with much less reliance of operator skill, as the transducer would not have to be held still for long periods of time, and the relatively short plane wave insonation minimizes artifacts due to respiratory and abdominal movement. A recent study by Yang et al. [43] showed an alternative means of acquiring WSS values using plane wave acquisitions. In this study, a series of carotid artery phantoms with varying degree of stenosis were analysed using a plane wave Doppler technique. Similarly to this work, the received signal was first demodulated in IQ components, followed by beamforming and filtering steps, and the velocity components were derived via an autocorrelation algorithm. The authors acknowledge a limitation of this method is the reduced SNR seen typically in all plane wave ultrasound acquisitions. They further state that attempting to improve the SNR via reduced pulse cycles or increased transmission frequency would result in a reduced velocity estimation accuracy. This is precisely the problem which is solved using the MFUDSA algorithm.

The MFUDSA algorithm consistently outperformed the other techniques assessed with the highest degree of significance detected out of the techniques tested and clearly delineating each of the vessel stiffnesses according to the different categories of disease progression. It is likely that a similar improvement to the PW measurements would be demonstrated in the pulsatile flow regime when using the MFUDSA algorithm, as found in the steady flow regime. This would allow for a more precise measurement to be made while significantly decreasing the scan time and lessening the influence of user proficiency.

A limitation of the measurements made with the string phantom in this current study was imposed by the maximum data length which could be recorded by the research mode on the Aixplorer system. This limit of 2000 data points made it impossible to generate a coherent Doppler spectrum for the experimental data, resulting in no discernible waveform for the measured data as shown in Figure 5a. The simulated data, on the other hand, did not have this limitation (Figure 4), and they behaved as expected. To verify this was the cause, another sonogram, 2000 points in length, was created from simulated data (Figure 5b). The data in Figure 5b demonstrate that, at shorter data lengths, considerable aliasing and loss of velocity information was evident.

A potential solution to this issue would be acquiring multiple acquisitions through the use of triggering, with new acquisitions offset by the duration of the previous acquisition. However, if this procedure was implemented, it may still be difficult to concatenate the resulting data files as they are not in a standard text readable format. These difficulties would be alleviated by implementing the analyses within the scanner itself due to considerable data memory savings this would allow.

## 5. Conclusions

The MFUDSA algorithm described herein demonstrates accurate determination of the quantitative WSS, which in turn can be used to estimate a patient’s risk for developing CVD. Earlier diagnosis of CVD is critical and modern gold standard techniques rely on the inference of arterial stenosis. Therefore, any technique which can indicate risk of CVD prior to stenosis formation would be of significant clinical value, as it can provide dramatically safer treatment options for patients who would otherwise have to undergo surgery when the disease has progressed further. The next step in this research is to evaluate our technique in vivo and compare it with existing methods for a better understanding of its role in early detection of cardiovascular disease.

## Figures and Tables

**Figure 1 diagnostics-13-01872-f001:**
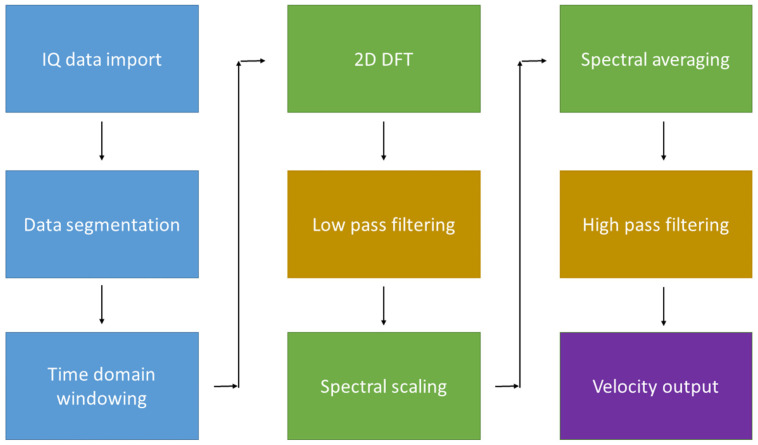
A block diagram of the steps in the MFUDSA algorithm. The diagram is colour-coded to correspond to the different stages in the cycle: blue items are processing operations taken on the data in the time domain; green items are processing steps taken in the frequency domain; yellow items are signal filtering; and purple corresponds to the output information sent to the user.

**Figure 2 diagnostics-13-01872-f002:**
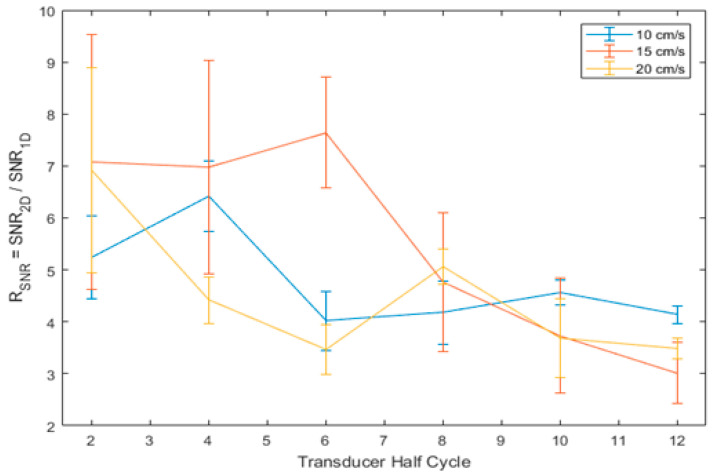
Variation of the ratio of two-dimensional SNR to one-dimensional SNR with respect to pulse length measured for three velocities. The error bars correspond to one standard deviation of 45 measurements for each measurement point.

**Figure 3 diagnostics-13-01872-f003:**
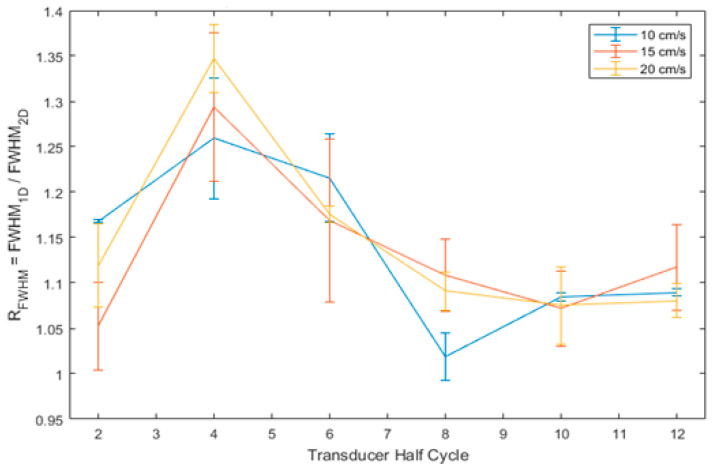
Variation of the ratio of one-dimensional velocity resolution (FWHM) to two-dimensional velocity resolution with respect to pulse length. The error bars correspond to one standard deviation.

**Figure 4 diagnostics-13-01872-f004:**
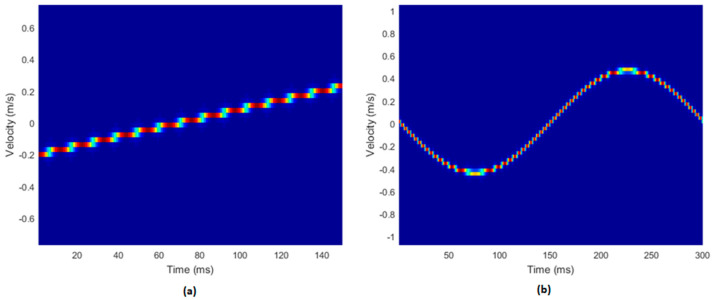
The simulated datasets. (**a**) The linearly varying velocity signal. The data length was 15,000, and the PRF was 5 kHz. (**b**) The cosine varying velocity signal. The data length was 21,000, and the PRF was 7 kHz.

**Figure 5 diagnostics-13-01872-f005:**
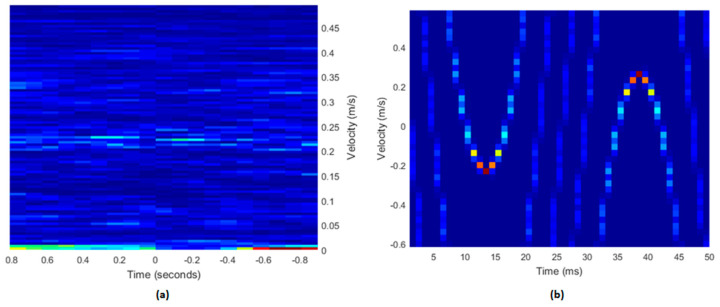
Sonogram outputs for experimental data. (**a**) The Doppler string phantom data. The data length was 2000, and the PRF was 4 kHz. (**b**) The cosine varying simulated dataset from Figure 4b with reduced data length. The data length was 2000, and the PRF was 4 kHz.

**Figure 6 diagnostics-13-01872-f006:**
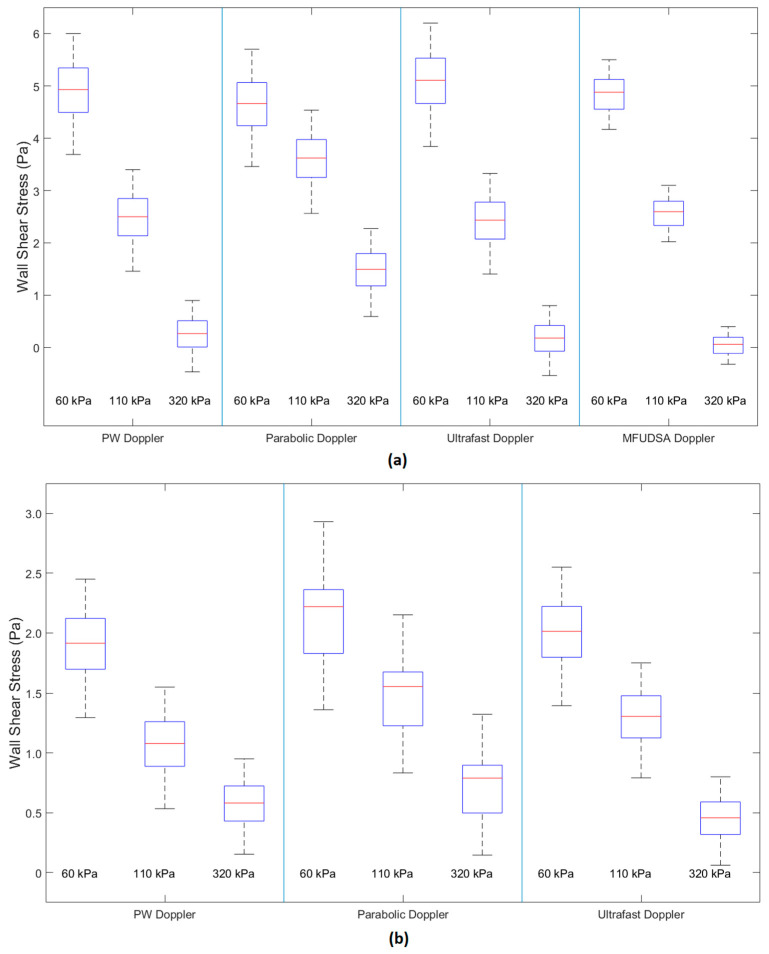
Boxplots of WSS values for a range of vessel stiffnesses in the (**a**) steady (n = 45) and (**b**) pulsatile flow (n = 30) regime grouped by estimation technique.

**Table 1 diagnostics-13-01872-t001:** Significance of the change in WSS with respect to vessel stiffness (*p* < 0.05).

Estimation Technique	Low to Medium Stiffness (Steady Flow)	Low to High Stiffness (Steady Flow)	Low to Medium Stiffness (Pulsatile Flow)	Low to High Stiffness (Pulsatile Flow)
PW Doppler	0.045	0.032	0.084	0.041
Parabolic Doppler	0.161	0.041	0.201	0.062
Ultrafast Doppler	0.051	0.039	0.097	0.032
MFUDSA Doppler	0.003	0.001		

## Data Availability

Data is available by contacting the corresponding author.

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
