# Peer review of "Development and Evaluation of a Multifrequency Ultrafast Doppler Spectral Analysis (MFUDSA) Algorithm for Wall Shear Stress Measurement: A Simulation and In Vitro Study"

_diagnostics, 2023, doi:10.3390/diagnostics13111872_

Round 1

Reviewer 1 Report

The article titled "Development and Evaluation of a Multifrequency ultrafast 2 Doppler spectral analysis (MFUDSA) algorithm for the early detection of cardiovascular disease: a simulation and in-vitro study" reports the development and testing of the Multifrequency ultrafast Doppler spectral analysis (MFUDSA). Reconstructions are limited to in-silico experiments with Doppler string phantom and in-vitro experiments with a standard flow phantom.

Authors compare the proposed MFUSDA technique against standard Doppler modes like PW, Ultrafast and Parabolic Doppler.

Overall, the article is well-written and of interest to this community. The article may be improved by considering the following comment:

1.     I would suggest the authors omit the phrase in the title:” for the early detection of cardiovascular disease”, since current study does not support the statement and it only reports feasibility in-silico and ex-vivo.

Author Response

We would like to thank the reviewer for the comments and we have changed the title of the manuscript as suggested.

Title changed to "Development and evaluation of a multifrequency ultrafast Doppler spectral analysis (MFUDSA) algorithm for wall shear stress measurement: a simulation and in vitro study"

Reviewer 2 Report

The work titled "Development and Evaluation of a Multifrequency ultrafast 2 Doppler spectral analysis (MFUDSA) algorithm for the early 3 detection of cardiovascular disease: a simulation and in-vitro 4 study" is well structured and written. It collects the most relevant antecedents and provides new knowledge that can be translated into a clinically useful tool in the near future that helps in the early diagnosis of CVD.

I encourage authors to study this tool in vivo and compare it with existing methods for a better understanding of its usefulness and contributions.

Author Response

We would like to thank the reviewer for the comments and agree that an evaluation of the in vivo performance is the next step.

We added the following sentence to the Conclusions

"The next step in this research is to evaluate our technique in-vivo and compare it with existing methods for a better understanding of it's role in the early detection of cardiovascular disease."

Reviewer 3 Report

Dear Editor.

The study focused on a novel algorithm for quantifying wall shear stress (WSS) in ather-22 osclerotic plaque using diagnostic ultrasound imaging, called Multifrequency ultrafast Doppler 23 spectral analysis (MFUDSA). The manuscript is well written, but I felt that the study would have been much better if it had the application such as invivo studies.

Please see minor comments below.

Line 18:  Cardiovascular pathology is global threat and a leading cause of death and disability, rephrase line 18.

Line 19: I cannot follow maybe mention name of the diagnostic testing method.

Line 29:  Define SNR

Line 31: What are other strategies? Is this a comparative study?

Line 78 to 81: Does this method improves the SNR?

Figure 6: Based on the results reported in figure 6. There is no significant improvement in Ultrafast Doppler compared to parabolic doppler and PW doppler. What is your conclusion based on these results?

It is not clear how MFUDSA has a better SNR compared to the other existing techniques.

Line 145: This manuscript remains a good idea. It would have been much better if it was inclusive of in vivo studies at least animal model.

References: Please ensure that all references consistence and formatted as per journal instructions.

Author Response

Reviewer 3

The study focused on a novel algorithm for quantifying wall shear stress (WSS) in atherosclerotic plaque using diagnostic ultrasound imaging, called Multifrequency ultrafast Doppler spectral analysis (MFUDSA). The manuscript is well written, but I felt that the study would have been much better if it had the application such as invivo studies.

Please see minor comments below.

Line 18:  Cardiovascular pathology is global threat and a leading cause of death and disability, rephrase line 18.

Line 18 rewritten as follows “Cardiovascular pathology is the leading cause of death and disability in the Western world and current diagnostic testing usually evaluates either the anatomy of the vessel, to determine if the vessel contains blockages and plaques.

Line 19: I cannot follow maybe mention name of the diagnostic testing method.

Line 19 rewritten as follows “However, there is a growing school of thought that other measures such as wall shear stress provide more useful information for earlier diagnosis and prediction of atherosclerotic related disease compared to pulsed-wave Doppler ultrasound, magnetic resonance angiography or computed tomography angiography.”

Line 29:  Define SNR – included in the abstract signal-to-noise ratio (SNR)

Line 31: What are other strategies? Is this a comparative study?

The following paragraph was included “The presented algorithm is compared with commonly used WSS assessment methods, such as, standard PW Doppler, Ultrafast Doppler and Parabolic Doppler as well as plane wave Doppler. Compared to an equivalent processing architecture with 1D Fourier analysis, the MFUDSA algorithm provided an increase in signal-to-noise ratio (SNR) by a factor of 4–8 and an increase in velocity resolution by a factor of 1.10–1.35. The results indicated that MFUDSA outperformed the other with significant differences detected between the typical WSS values of moderate disease progression (p=0.003) and severe disease progression (p=0.001). The algorithm demonstrated an improved performance for the assessment of WSS and has potential to provide an earlier diagnosis of cardiovascular disease than current techniques allow.”

Line 78 to 81: Does this method improves the SNR?

As mentioned in the text, this method only improves temporal resolution, SNR remains the same.

Figure 6: Based on the results reported in figure 6. There is no significant improvement in Ultrafast Doppler compared to parabolic doppler and PW doppler. What is your conclusion based on these results?

Ultrafast Doppler is not intended to demonstrate an improvement over the other technologies. Here, Ultrafast Doppler is identical to the MFUDSA technique without applying the 2D Fourier technique, instead using a 1D technique. The column for the MFUDSA does show a significant difference, therefore illustrating that the observed improvements from the MFUDSA algorithm come from the 2D analysis and not only from the Ultrafast acquisition method.

It is not clear how MFUDSA has a better SNR compared to the other existing techniques.

Figure 6 does not contain information on SNR, Figure 2 shows the ratio of SNR using 2D analysis to using 1D analysis and illustrates that a reduction in transducer half-cycle improves SNR in 2D analysis relative to 1D.

Line 145: This manuscript remains a good idea. It would have been much better if it was inclusive of in vivo studies at least animal model.

Response – We agree that including in vivo data would improve this study but it was outside the scope of this work, we added the following sentence to the conclusions: “The next step in this research is to evaluate our technique in-vivo and compare it with existing methods for a better understanding of it role in early detection of cardiovascular disease.”

References: Please ensure that all references consistence and formatted as per journal instructions.

References are using Multidisciplinary Digital Publishing Institute style as specified in the instructions for authors section.